# Multiscale Entropy Analysis of Short Signals: The Robustness of Fuzzy Entropy-Based Variants Compared to Full-Length Long Signals

**DOI:** 10.3390/e23121620

**Published:** 2021-12-01

**Authors:** Airton Monte Serrat Borin, Anne Humeau-Heurtier, Luiz Eduardo Virgílio Silva, Luiz Otávio Murta

**Affiliations:** 1Federal Institute of Education, Science and Technology of Triangulo Mineiro, Uberaba 38064-790, Brazil; airton@iftm.edu.br; 2LARIS—Laboratoire Angevin de Recherche en Ingénierie des Systèmes, University of Angers, 49035 Angers, France; anne.humeau@univ-angers.fr; 3Department of Internal Medicine, Ribeirão Preto Medical School, University of São Paulo, Ribeirão Preto 14049-900, Brazil; luizeduardo@usp.br; 4Department of Computing and Mathematics, School of Philosophy, Sciences and Languages of Ribeirão Preto, University of São Paulo, Ribeirão Preto 14040-901, Brazil

**Keywords:** multiscale fuzzy entropy, time series

## Abstract

Multiscale entropy (MSE) analysis is a fundamental approach to access the complexity of a time series by estimating its information creation over a range of temporal scales. However, MSE may not be accurate or valid for short time series. This is why previous studies applied different kinds of algorithm derivations to short-term time series. However, no study has systematically analyzed and compared their reliabilities. This study compares the MSE algorithm variations adapted to short time series on both human and rat heart rate variability (HRV) time series using long-term MSE as reference. The most used variations of MSE are studied: composite MSE (CMSE), refined composite MSE (RCMSE), modified MSE (MMSE), and their fuzzy versions. We also analyze the errors in MSE estimations for a range of incorporated fuzzy exponents. The results show that fuzzy MSE versions—as a function of time series length—present minimal errors compared to the non-fuzzy algorithms. The traditional multiscale entropy algorithm with fuzzy counting (MFE) has similar accuracy to alternative algorithms with better computing performance. For the best accuracy, the findings suggest different fuzzy exponents according to the time series length.

## 1. Introduction

Complex systems are composed of many agents interacting with each other by nonlinear rules and exhibiting temporal and spatial structures at different scales [1]. Quantifying the complexity level from realizations of the system’s dynamics, i.e., time series, is still a challenge. While different interpretations of complexity may be assumed, entropy certainly plays a role in the estimation of time series complexity [2,3].

Most entropy estimators for time series are inspired by the Kolmogorov-Sinai (KS) entropy, e.g., sample entropy [4] and fuzzy entropy [5]. They intend to estimate the rate at which the information grows over time in the system. However, the introduction of multiscale entropy (MSE) [6] was a milestone in the field of complexity analysis since the multiscale aspects of the system’s dynamics can now be taken into account. The MSE algorithm is based on a coarse-graining procedure for generating the scaled versions of the original dynamics followed by the calculation of sample entropy for each scaled series.

Although MSE showed itself worthful in discriminating different complex dynamics, it introduces bias when dealing with short-term time series. First, the coarse-graining procedure of MSE drastically decreases the time series’s length, decreasing the number of points available for entropy estimation. Second, the sample entropy algorithm is based on similar pattern counting, and short time series may result in a biased or even undefined value of entropy. MSE proves to be inaccurate in short time series analysis [7,8], significantly losing its sensitivity [9]. To overcome these limitations, some approaches propose different coarse-graining procedures and entropy estimation. Composite MSE (CMSE) [10], refined composite MSE (RCMSE) [11], and modified MSE (MMSE) [7,12] are important examples. Multiscale fuzzy entropy (MFE) uses a fuzzy membership function to identify similarities between patterns within time series, avoiding zero counts or numeric instabilities in entropy calculation [13,14,15].

Some studies also combined the advantages of improved coarse-graining procedures with the fuzzy entropy estimation. Composite and refined composite multiscale fuzzy entropy (CMFE and RCMFE) present a joint approach with CMSE and RCMSE, respectively. Both CMFE and RCMFE were evaluated in the biomedical and non-biomedical contexts [15,16,17,18]. In a recent study, we proposed and evaluated the modified multiscale fuzzy entropy (MMFE) for heart rate variability (HRV) analysis [19]. In a systematic comparison, we showed that MMFE is more robust for estimating original MSE than MMSE when the fuzzy parameter is optimized.

This study systematically compares the accuracy of CMSE, RCMSE, MMSE, MFE, CMFE, RCMFE, and MMFE to estimate the original MSE in a short time series. The study considered MSE estimated for full length series as reference to calculate the errors for each method variations. Each method accuracy is analyzed for different lengths of series, and the dependence of the best fuzzy exponent on the series length is reported. We seek to find the most accurate and cost-effective multiscale entropy measure for short time series.

## 2. Materials and Methods

We handled experiments with long HRV time series obtained from two biological databases (rats and humans) and exhaustively applied CMSE, RCMSE, MMSE, MFE, CMFE, RCMFE, and MMFE to different sizes of segments. The minimum error compared to the original long-term MSE was considered as the maximum accuracy. The corresponding multiscale entropy algorithms are briefly detailed in the following subsections.

For all methods, consider a time series **u** with *N* samples defined as *u(1)*, *u(2)*, …, *u(N)*. We define *m* as the length of vectors (patterns) to be compared and *r* as the tolerance accepted between corresponding points within the vectors. This tolerance is defined as a percentage of the original time series SD. There are N−mδ+1 template vectors **x**${}_{m}\left(i\right)$ for {i∣1≤i≤N−mδ+1}, where **x**${}_{m}\left(i\right)=\left\{u\left(i+k\delta \right):0\leq k\leq m-1\right\}$ is a vector with length *m* and δ is the delay considered between samples.

### 2.1. Sample Entropy (SampEn) and Fuzzy Entropy (FuzzyEn)

The sample entropy (*SampEn*) algorithm [4,20] and the fuzzy entropy (*FuzzyEn*) algorithm [5] calculate the distance between any two vectors as
(1)$$d\left[x_{m}\left(i\right),x_{m}\left(j\right)\right]=\max_{0\leq k\leq m-1}\left|u\left(i+k\delta \right)-u\left(j+k\delta \right)\right|,$$
where j>i+δ.

#### 2.1.1. SampEn

First, $B_{i}$ is calculated as the number of matches for the template vector $x_{m}\left(i\right)$, i.e., the number of vectors $x_{m}\left(j\right)$ which distances $d[x_{m}\left(i\right),x_{m}\left(j\right)]$ are less than or equal to *r*, for 0≤j≤N−mδ. Next, $A_{i}$ is calculated as the number of matches for the template vector $x_{m+1}\left(i\right)$, i.e., the number of vectors $x_{m+1}\left(j\right)$ which the distances $d[x_{m+1}\left(i\right),x_{m+1}\left(j\right)]$ are less than or equal to *r*, for 0≤j≤N−mδ. Then, $C^{m}\left(r\right)$ and $C^{m+1}\left(r\right)$ are computed as
(2)$$C_{i}^{m}=\frac{B_{i}}{N-m\delta -1},$$
(3)$$C^{m}\left(r\right)=\frac{1}{N-m\delta }\sum_{i=1}^{N-m\delta }C_{i}^{m}\left(r\right),$$
and
(4)$$C_{i}^{m+1}=\frac{A_{i}}{N-m\delta -1},$$
(5)$$C^{m+1}\left(r\right)=\frac{1}{N-m\delta }\sum_{i=1}^{N-m\delta }C_{i}^{m+1}\left(r\right).$$

Then, *SampEn* is obtained as the negative logarithm of the conditional probability $C^{m+1}\left(r\right)/C^{m}\left(r\right)$, estimated with the parameters *m*, *r*, and δ:(6)$$SampEn\left(u,m,r,\delta \right)=-\ln\frac{C^{m+1}\left(r\right)}{C^{m}\left(r\right)}$$

#### 2.1.2. FuzzyEn

*FuzzyEn* is based on the concept of fuzzy sets [21], defining the similarity levels between vectors by the fuzzy associative (membership) function and the vectors’ distances. Vectors $x_{m}\left(i\right)$ are created similarly to *SampEn*, except by the fact that the mean vector baseline is removed:(7)$$x_{m}\left(i\right)=\left\{u\left(i+k\delta \right)-u0\left(i\right):\;0\leq k\leq m-1\right\}$$
where
(8)$$u0\left(i\right)=\frac{1}{m}\sum_{j=0}^{m-1}u\left(i+j\delta \right).$$

To calculate the similarity between two vectors, two functions were tested in our work: $\exp(-d_{m}^{n}/r)$ and $\exp(-0.6931{\left(d/r\right)}^{n})$ [22]. For the first function, we computed
(9)$$B_{ij}^{m}\left(r\right)=\exp\left(-d_{m}^{n}/r\right)$$
and
(10)$$A_{ij}^{m}\left(r\right)=\exp\left(-d_{m+1}^{n}/r\right),$$
where *d* is given in Equation (Equation 1) and *n* is the exponent of the fuzzy function. For the second function, $B_{ij}^{m}\left(r\right)$ and $A_{ij}^{m}\left(r\right)$ were computed similarly but using $\exp(-0.6931{\left(d/r\right)}^{n})$.

We also define
(11)$$\phi^{m}\left(n,r\right)=\frac{1}{N-m\delta }\sum_{i=1}^{N-m\delta }B_{ij}^{m}\left(r\right).$$
and
(12)$$\phi^{m+1}\left(n,r\right)=\frac{1}{N-m\delta }\sum_{i=1}^{N-m\delta }A_{ij}^{m+1}\left(r\right),$$
similar to Equations (Equation 3) and (Equation 5), so *FuzzyEn* for the parameters *m*, *n*, δ, and *r* is calculated by
(13)$$FuzzyEn\left(u,m,n,r,\delta \right)=-\ln\frac{\phi^{m+1}\left(n,r\right)}{\phi^{m}\left(n,r\right)}.$$

### 2.2. Multiscale Entropy (MSE) and Multiscale Fuzzy Entropy (MFE)

In the *MSE* [6,8,23] and *MFE* [13,14,15] algorithms, the dynamics of a system at different time scales is obtained by a moving average procedure (*coarse-graining* procedure), according to
(14)$$u^{\tau }\left(j\right)=\frac{1}{\tau }\sum_{i=(j-1)\tau +1}^{j\tau }u\left(i\right),\;1⩽j⩽N/\tau .$$

The irregularities in the time series for the scale factor τ are quantified by applying *SampEn* (*FuzzyEn*) for *MSE* (*MFE*) on the coarse-grained time series obtained, with unitary delay (δ=1), that is
(15)$$MSE\left(u,m,r\right)=SampEn\left(u^{\tau },m,r,\delta =1\right)$$
and
(16)$$MFE\left(u,m,n,r\right)=FuzzyEn\left(u^{\tau },m,n,r,\delta =1\right).$$

### 2.3. Composite Multiscale Entropy (CMSE) and Composite Multiscale Fuzzy Entropy (CMFE)

In *CMSE* [10] and *CMFE* [15], for each scale factor τ, *k**coarse-grained* time series **y**${}_{k}^{\tau }$ are obtained where the elements of the *k*-th series are defined as
(17)$$y_{k}^{\tau }\left(j\right)=\frac{1}{\tau }\sum_{i=(j-1)\tau +k}^{j\tau +k-1}u_{i},\;\;1\leq j\leq \frac{N}{\tau },\;1\leq k\leq \tau .$$

*CMSE* and *CMFE* at scale τ are defined as the average entropy obtained from the *k* series at scale τ, that is
(18)$$CMSE\left(u,\tau ,m,r\right)=\frac{1}{\tau }\sum_{k=1}^{\tau }SampEn\left(y_{k}^{\tau },m,r,\delta =1\right)$$
and
(19)$$CMFE\left(u,\tau ,m,r\right)=\frac{1}{\tau }\sum_{k=1}^{\tau }FuzzyEn\left(y_{k}^{\tau },m,n,r,\delta =1\right).$$

### 2.4. Refined Composite Multiscale Entropy (RCMSE) and Refined Composite Multiscale Fuzzy Entropy (RCMFE)

The procedure to obtain the *coarse-grained* series in *RCMSE* [11] and *RCMFE* [24] is the same as for *CMSE* and *CMFE* (see Equation (Equation 17)). However, instead of averaging the entropy of each *k* scaled series for scale τ, entropy is estimated from the average number of matches $n_{k,\tau }^{m+1}$ and $n_{k,\tau }^{m}$, obtained from all *k* *coarse-grained* series.

#### 2.4.1. RCMSE

*RCMSE* is defined as:(20)$$RCMSE\left(u,\tau ,m,r\right)=-\ln\left(\frac{{\bar{n}}_{k,\tau }^{m+1}}{{\bar{n}}_{k,\tau }^{m}}\right),$$
where ${\bar{n}}_{k,\tau }^{m+1}=\frac{1}{\tau }\sum_{k=1}^{\tau }n_{k,\tau }^{m+1}$ and ${\bar{n}}_{k,\tau }^{m}=\frac{1}{\tau }\sum_{k=1}^{\tau }n_{k,\tau }^{m}$ are the averages of $n_{k,\tau }^{m+1}$ and $n_{k,\tau }^{m}$ respectively.

#### 2.4.2. RCMFE

Given ${\bar{\phi }}_{\tau }^{m+1}$ and ${\bar{\phi }}_{\tau }^{m}$ as the averages of $\phi^{m+1}$ and $\phi^{m}$ for each *k* at the scale factor τ, respectively, *RCMFE* is defined by:(21)$$RCMFE\left(u,\tau ,m,n,r\right)=-\left(\ln\frac{{\bar{\phi }}_{k,\tau }^{m+1}}{{\bar{\phi }}_{k,\tau }^{m}}\right),$$
where $\phi^{m}$ and $\phi^{m+1}$ are given by Equations (Equation 11) and (Equation 12), respectively.

### 2.5. Modified Multiscale Entropy (MMSE) and Modified Multiscale Fuzzy Entropy (MMFE)

In *MMSE* and *MMFE*, the scaled versions **z**${}^{\tau }$ of the original time series are created according to the following procedure:(22)$$z^{\tau }\left(i\right)=\frac{1}{\tau }\sum_{k=i}^{i+\tau -1}u\left(k\right),\;1⩽i⩽N-\tau +1,$$
where τ represents the time scale factor. This procedure is similar to the one adopted in *MSE* (Equation (Equation 14)), except for the overlapping in the moving average. In this procedure, the length of the *coarse-grained* time series obtained using the overlapping moving average for a scalar of τ is N−τ+1, remarkably greater compared to the length of the *coarse-graining* procedure of *MSE* (N/τ).

#### 2.5.1. MMSE

The *MMSE* method [10] proposes that the *coarse-grained* time series be constructed by Equation (Equation 22) and that the entropy is estimated for each scale factor τ by applying *SampEn* with a time delay equal to τ, that is: (23)$$MMSE\left(u,m,\tau ,r\right)=SampEn\left(z^{\tau },m,r,\delta =\tau \right).$$

#### 2.5.2. MMFE

MMFE was recently proposed [19] and consists in the application of the same *coarse-graining* procedure as *MMSE* (Equation (Equation 22)) followed by the estimation of entropy using a delayed version of *FuzzyEn*, with a delay equal to τ for each scale. The equation of MMFE is given by:(24)$$\mathrm{MMFE}\left(u,n,m,\tau ,r\right)=FuzzyEn\left(z^{\tau },m,n,r,\delta =\tau \right).$$

## 3. Dataset and Experiments

### 3.1. Dataset

Heart rate variability (HRV) series from rats and humans were obtained from previous studies [19,25]. The first group of ECG data was recorded in 18 healthy Wistar rats. The recordings were performed in the Cardiovascular Physiology Laboratory of Ribeirão Preto Medical Schools, University of São Paulo. Briefly, the rats had their ECG recorded for approximately 1 h (40 to 80 min) at baseline conditions. Computer software (LabChart, ADInstruments, Sydney, Australia) was used to create RR series from ECG recordings, sampled at 2 kHz. All RR series were visually inspected for artifacts and corrected when necessary. Since the time series’s length varied from 15,892 to 32,333 points, all RR series were truncated to 15,892 points. The second group of HRV series consisted of 12 healthy human individuals, obtained from the Physionet MIT-BIH Normal Sinus Rhythm database digitized at 360 Hz per signal lead [26]. The 12 ECG recordings were selected randomly from the database. The RR series were calculated using the *ann2rr* tool from the WFDB Physionet package, which uses the recordings’ beat annotations to calculate the RR intervals. Only normal RR intervals were considered, that is, the intervals between two successive normal beats. Eventually, all RR series were truncated to 15,892 samples so that the series of rats and humans had the same length. The recording period of all series ran from 8 a.m. to 10 p.m.

### 3.2. Experiments

We segmented the HRV full series (15,892 points) into equal segments of 400, 800, 1200, …, 15,600 points, with a superimposition of 90% to the previous segment. For each segment, all the variants of MSE described above were computed, and the average value over the segments with the same size was taken to represent the whole series. The maximum scale factor assessed was 20, i.e., τ=1,2,…,20. The embedding dimension and tolerance factor of entropy estimators were set as m=2 and r=0.15× SD of the series, respectively. To evaluate each MSE variant’s accuracy on the estimation of the original MSE, the mean square error was calculated over all time scales, always taking the original MSE, obtained from the full-length series, as reference. The error was calculated for each series and each segment size, and the mean errors were reported as a function of the segment size.

The mean squared error is obtained by calculating the entropy for time scales from 1 to 20 for each segment of 400, 800, …, 15,600 points. If the number of windows is greater than 1, the arithmetic average of each scaling factor is made. For example: for the 400-point segment, we have 385 windows. For each window, we calculated entropy on the 20 time scales. Then, the arithmetic mean of the 385 windows is taken for each scaling factor, so the entropy for scale 1 and the 400-point segment is the entropy average for scale 1 of all 385 windows. The entropy for scale 2 and the segment of 400 points is the average entropy on scale 2 of all 385 windows, so up to scale 20. As a result, we have an average entropy for scales 1, 2, …, 20. We then average the mean squared error of these entropy values with the calculated MSE entropy on a scale of 1 to 20.

Moreover, to assess the cost-effectiveness of fuzzy-based MSE variants, we measured each algorithm’s average computation time. The analysis was performed on a desktop computer equipped with an Intel Core i7 930@2.8 GHz processor and 16 Gb of RAM. To guarantee the isonomy of the results, all the tests were performed with the MATLAB software (The MathWorks, Inc. Natick, MA, USA) and the maxNumCompThreads=1 command so that all methods used a single CPU. The average time consumed to process three randomly selected human HRV series is reported as a segment size function (from 400 to 12,000 points). The fuzzy exponent *n* adopted in this experiment followed the equation previously found for the choice of the best exponent according to the segment size (*x*) [19]:(25)$$n=0.82+0.10\exp(-3\times x/10^{4}).$$

## 4. Results

The accuracy of CMSE, RCMSE, MMSE, CMFE, RCMFE, MMFE, and MFE were evaluated as the error compared to the MSE calculated using the full-length series. Figure 1 shows the accuracy of MFE obtained with both rats and humans HRV series. The descriptive measures of the MSE for both the human and animal datasets can be found in a previous paper [19]. The top left plot shows the mean squared errors as a function of the rat dataset’s segment size. We illustrate different error curves for different fuzzy exponents ranging from n=0.8 to n=1.5. The magnification of the errors’ curves for short segment sizes are shown at the bottom left corner, and the minimum error for each segment size is shown on the right side of the magnification plot. The mean squared errors are shown as a segment size function for the humans’ dataset at the top right corner. We illustrate different error curves for different fuzzy exponents ranging from n=0.85 to n=0.92. One can find the magnification of the errors’ curves for short segment sizes at the bottom right corner, as well as the minimum errors obtained for each segment size.

As can be seen in Figure 1, the error for each fuzzy exponent *n* depends on the segment size (series length), and the optimal *n* are the ones that provide the lowest errors. For the HRV series from rats, the best exponents increase with the segment size, while it decreases for human HRV series.

Figure 2 shows the mean squared errors for CMSE (dashed line) and CMFE (solid lines). Error curves for CMFE are illustrated for fuzzy exponents ranging from n=0.8 to n=1.5 (rats) and n=0.85 to n=0.92 (humans), similar to Figure 1. At the bottom plots, one can see the magnifications of error curves for short segment sizes and the minimum errors for CMFE (black line) compared to CMSE (gray line). Similar to the results with MFE, the errors are dependent on the segment size. The best exponents increase with the segment size for rats while it decreases for human HRV series.

Figure 3 presents the error curves for RCMSE (dashed line) and RCMFE (solid lines). Error curves are illustrated for the fuzzy exponents ranging from n=0.8 to n=1.5 (rats) and n=0.85 to n=0.92 (humans). Magnification of the error curves and the minimum errors for all segment sizes are shown at the bottom plots. Like MFE and CMFE, the errors of RCMFE are dependent on the segment size, and the best exponents increase with the segment size for rats, while it decreases for human HRV series.

The error curves obtained with MMFE and MMSE are shown in Figure 4. Although a similar error plot can be found in a previous study [19], here we expanded the range of exponents evaluated to calculate the minimum MMFE error. Error curves are illustrated for the fuzzy exponents ranging from n=0.8 to n=1.5 (rats) and n=0.85 to n=0.92 (humans). Magnification of the error curves, together with plots of the minimum MMFE and MMSE errors for all segment sizes, are shown at bottom plots. Like all the other fuzzy-based MSE variants, the errors of MMFE depend on the segment size, and the best exponents increase with the segment size for rats, while it decreases for human HRV series.

Figure 5 compares errors from all the multiscale variants studied, i.e., CMSE, RCMSE, MMSE, CMFE, RCMFE, MMFE, and MFE. The minimum error is shown for fuzzy entropy-based methods, calculated with each segment size’s optimal fuzzy exponent. The figure shows that all variants based on diffuse entropy have fewer errors than any variant based on sample entropy. For segments sized up to 13,000 points, the MFE, CMFE, and RCMFE curves are superimposed because these methods have similar results. Mean squared error for all considered approaches, i.e., CMSE, RCMSE, MMSE, CMFE, RCMFE, MMFE, and MFE. For fuzzy entropy-based approaches, only the minimum error is shown, obtained with the optimal fuzzy exponent for each segment size. Results are shown for both rats (left) and human (right) database. Note that the errors are calculated regarding the MSE of full-length time-series, i.e., 15,892 beats.

Figure 6 shows the optimal fuzzy exponents of fuzzy-based MSE variants for each segment size. These exponents were utilized to calculate the minimum error curves for multiscale fuzzy entropy-based variants (see Figure 5). The curves were fitted to exponential functions, which can be employed to find the best fuzzy exponent of those datasets according to the time series length. Note that for the HRV series from rats, the optimal fuzzy exponent increases with the series length, decreasing the HRV series of humans.

For the sake of comparison, Figure 7 shows the mean square error of MFE using the alternative fuzzy function $\exp(-0.6931\times {\left(d/r\right)}^{n})$. The errors are illustrated for the fuzzy exponent ranging from n=1.3 to n=3.0 (rats) and from n=2.0 to n=5.5 (humans). Results show that MFE with this alternative fuzzy function also presents dependence on the segment size, similar to the original fuzzy function (Figure 1). However, the values of optimal exponents for each segment size are markedly different from the original fuzzy function, and the best *n* increases with the segment size for both rats and humans HRV series.

Figure 8 shows the average time (over three human HRV series) spent calculating the different multiscale fuzzy entropy-based variants at increasing segment sizes (up to 12,000 samples). As expected, the computational time required to run any method increases with the segment size. However, MMFE has the highest computational cost, while MFE showed the lowest. CMFE and RCMFE show virtually the same computational time.

## 5. Discussion

In the present study, we adopted a systematic comparison between three variants of MSE (CMSE, RCMSE, MMSE) and their fuzzy-based adaptations (CMFE, RCMFE, MMFE, MFE) to check the accuracy of them to estimate the real MSE for short-term signals. As expected, all fuzzy-based methods performed superior compared to the algorithms based on sample entropy. Surprisingly, all fuzzy-based methods’ accuracy is quite similar, pointing that the use of fuzzy entropy in place of sample entropy seems sufficient to provide optimal estimations of MSE for short-term signals. In other words, the improvements adopted in the coarse-graining for CMFE, RCMFE, and MMFE seem to have little or no effect in fuzzy-based variants, since MFE provided errors in the estimate of original MSE as low as the ones found in CMFE, RCMFE, and MMFE. The replacement of the rigid similarity of SampEn (Heaviside function) by the smooth fuzzy function in FuzzyEn seems to be the most relevant improvement for a good estimation of entropy in short time series. However, one must be aware that optimal fuzzy exponents’ choice is crucial to obtain high accuracy.

In a previous study with the same dataset, we showed that MMFE provides better estimates of the original MSE than MMSE when proper choices of the fuzzy exponent *n* are made [19]. Here, we showed that CMFE, RCMFE, and MFE also have a dependence on *n* and that the optimal exponents found for MMFE are virtually the same for CMFE, RCMFE, and MFE, as can be seen in Figure 6. Although both the rats and humans datasets represent health conditions, the best exponents for rats increase with series length, while it decreases for humans. This is likely to be a consequence of the different species, but it still has to be investigated together with datasets with pathological HRV series. Nevertheless, the fitting equations provided in Figure 6 can be used for the choice of the optimal fuzzy exponents in the dataset evaluated here. The measure stationarity is a possible issue concerning long-term time-series as previously pointed out [27]. However, all time-series and analyzed segments are supposed to be taken at baseline physiological state. We introduce an illustrative analysis of two time-series data, i.e., one human and one rat, in the appendix to this paper presented in Figure A1.

To check the influence of the fuzzy function on the accuracy of fuzzy-based MSE variants, we calculated the mean squared error of MFE using an alternative fuzzy function, i.e., $\exp(-0.6931\times {\left(d/r\right)}^{n})$ [22]. Interestingly, this alternative function also presented a dependence of the minimum error on the choice of the exponent *n*. However, the range of optimal *n* values is markedly different from the ones found for the original fuzzy function, and curiously, the exponent always increases with the series length for both datasets (rats and humans). The extensive evaluation of different fuzzy functions is out of the present study’s scope, and one must be aware that changing the fuzzy function requires the search for the optimal fuzzy exponents. For an extensive review on the possible fuzzy functions and their differences, please refer to [28].

For entropy estimators based on a similarity function between patterns (such as SampEn and FuzzyEn), the tolerance factor, *r*, is commonly defined as a percentage of the signal’s SD, making the results comparable within signals with different magnitude. Alternatively, the signal can be normalized to mean zero and SD one, a procedure that has the same effect of multiplying the tolerance factor by the signal SD. However, in FuzzyEn, these two procedures are not always equivalent and depend on the fuzzy function adopted. In the case of $\exp(d^{n}/r)$ (the fuzzy function adopted in this study), one can notice that the distance between patterns (*d*) and *r* are not raised to the same power (except when n=1). Thus, normalizing the series (affecting *d*) or normalizing *r* may result in different entropy values. The alternative fuzzy function evaluated with MFE ($\exp(-0.6931\times {\left(d/r\right)}^{n})$) does not show this limitation, since *d* and *r* are both raised to *n*.

The computation cost (time) analysis necessary to calculate all the fuzzy-based variants of MSE showed that MMFE is the most time-consuming. On the other hand, MFE is the fastest algorithm among them. Since all algorithms’ accuracy is very similar, the simplicity and computational efficiency of MFE make this algorithm the most attractive to be used for the analysis of short-term signals, which provides good accuracy for HRV series as short as 400 points.

## 6. Conclusions

In this study, several SampEn- and FuzzyEn-based MSE variants were calculated in short HRV time series, and their accuracy to estimate the actual MSE was investigated. All fuzzy entropy-based algorithms provided better accuracy (given the fuzzy exponent’s proper choice) compared to the variants based on SampEn. Moreover, all FuzzyEn-based algorithms evaluated showed similar accuracy. Therefore, since MFE is the most simple and cost-effective algorithm among them, we recommend the use of MFE for the analysis of short-term time series. The results also indicate that different fuzzy functions may provide good accuracies. However, the dependence of the fuzzy exponent (*n*) to the series length may vary from one function to another and on different datasets. Further studies are necessary to determine the optimal fuzzy exponents in datasets with pathological signals. A possible limitation of the study is the fact of assuming the entropy computed for the whole series compared to shorter segments of itself, although the comparison can reflect accuracy and precision.

## Figures and Tables

**Figure 1 entropy-23-01620-f001:**
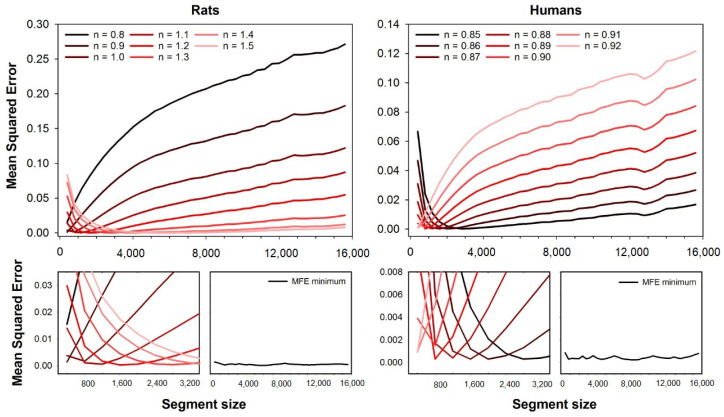
Mean squared errors of MFE as a function of the segment size. The errors were obtained for fuzzy exponents varying in the range n=0.8 to n=1.5 (rats) and n=0.85 to n=0.92 (humans). Top plots show all the error curves, while bottom plots show magnification of short segment sizes’ errors. One can notice that the fuzzy exponent that gives the best accuracy (lower error) varies according to the segment size. The lowest error for all segment sizes is illustrated in the plots at the magnification plots’ right (MFE minimum).

**Figure 2 entropy-23-01620-f002:**
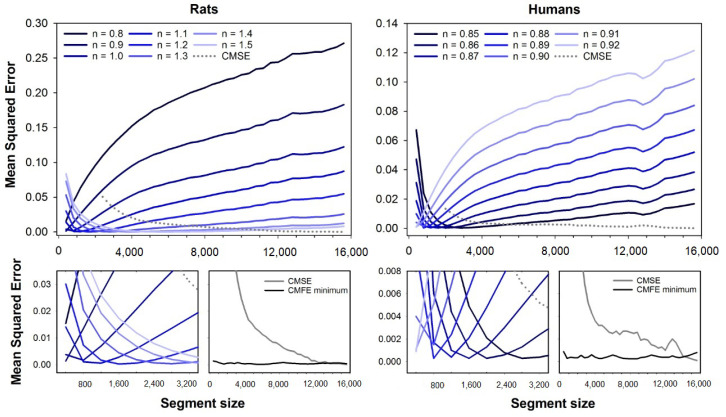
Mean squared errors of CMFE and CMSE as a function of the segment size. The errors of CMFE were obtained for fuzzy exponents varying in the range n=0.8 to n=1.5 (rats, at **left**) and n=0.85 to n=0.92 (humans, at **right**). **Top** plots show the full error curves of CMFE (solid lines) and CMSE (dashed line), while **bottom** plots show a magnification of the errors for short segment sizes. One can notice that the fuzzy exponent that gives the best accuracy (lower error) for CMFE varies according to the segment size. The lowest error for CMFE (black lines) and CMSE (gray lines) for all segment sizes are illustrated in the plots at the magnification plots’ right.

**Figure 3 entropy-23-01620-f003:**
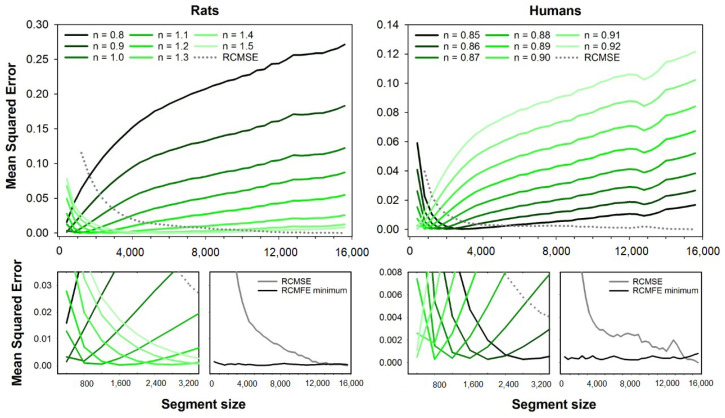
Mean squared errors of RCMFE and RCMSE as a function of the segment size. The errors of RCMFE were obtained for fuzzy exponents varying in the range n=0.8 to n=1.5 (rats, at **left**) and n=0.85 to n=0.92 (humans, at **right**). **Top** plots show the full error curves of RCMFE (solid lines) and RCMSE (dashed line), while **bottom** plots show magnification of short segment sizes’ errors. One can notice that the fuzzy exponent that gives the best accuracy (lower error) for RCMFE varies according to the segment size. The lowest error for RCMFE (black lines) and RCMSE (gray lines) for all segment sizes are illustrated in the plots at the magnification plots’ right.

**Figure 4 entropy-23-01620-f004:**
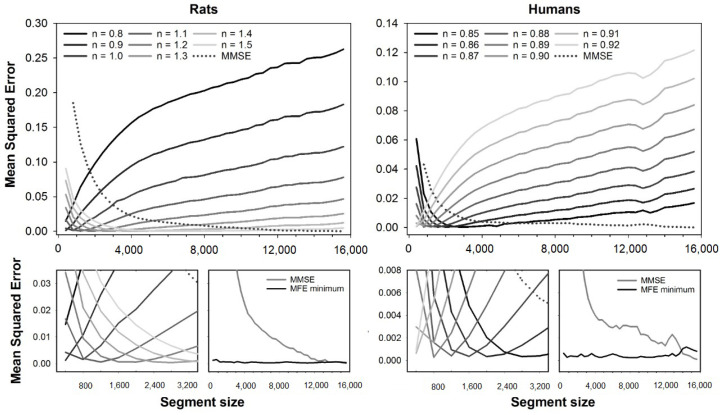
Mean squared errors of MMFE and MMSE as a function of the segment size. The errors of MMFE were obtained for fuzzy exponents varying in the range n=0.8 to n=1.5 (rats, at **left**) and n=0.85 to n=0.92 (humans, at **right**). **Top** plots show the full error curves of MMFE (solid lines) and MMSE (dashed lines), while **bottom** plots show a magnification of the errors for short segment sizes. One can notice that the fuzzy exponent that gives the best accuracy (lower error) for MMFE varies according to the segment size. The lowest error for MMFE (black lines) and MMSE (gray lines) for all segment sizes are illustrated in the plots at the magnification plots’ right.

**Figure 5 entropy-23-01620-f005:**
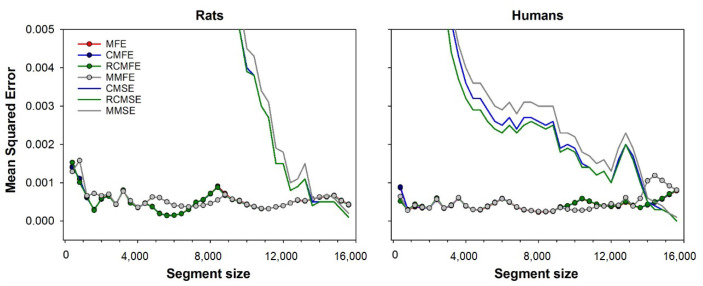
Mean squared error for all considered approaches, i.e., CMSE, RCMSE, MMSE, CMFE, RCMFE, MMFE and MFE. For fuzzy entropy-based approaches, only the minimum error is shown, obtained with the optimal fuzzy exponent for each segment size. Results are shown for both rats (**left**) and human (**right**) database. Notice that the errors are calculated regarding the MSE of full-length time-series, i.e., 15,892 beats. The figure shows that all variants based on diffuse entropy have fewer errors than any variant based on sample entropy. For segments sized up to 13,000 points, the MFE, CMFE, and RCMFE curves are superimposed because these methods have similar results.

**Figure 6 entropy-23-01620-f006:**
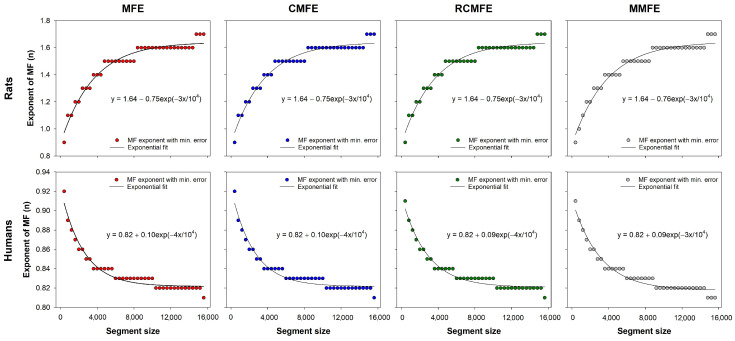
Best fuzzy exponents found for each segment size in both rats and humans HRV datasets. The solid lines represent the best fitting functions, i.e., a decreasing exponential function $y=1.64-0.75\exp(-3x/10^{4})$ for rats and an increasing exponential $y=0.82-0.10\exp(-4x/10^{4})$ for humans. These functions can be used to choose the best fuzzy exponent of these datasets according to the series length.

**Figure 7 entropy-23-01620-f007:**
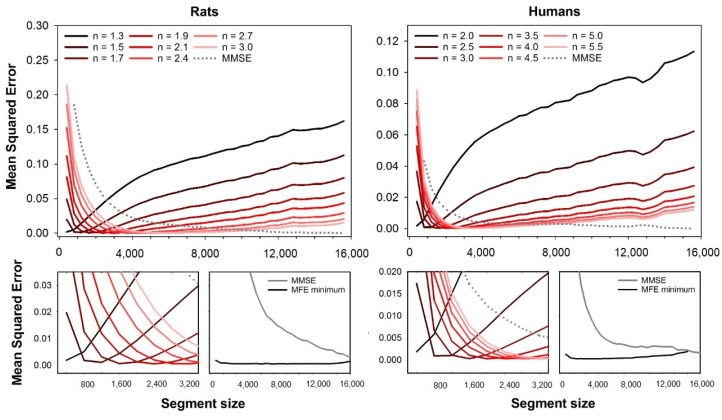
Mean squared errors of MFE as a function of the segment size, obtained with alternative fuzzy function, i.e., $\exp(-0.6931\times {\left(d/r\right)}^{n})$. The errors are illustrated for the fuzzy exponent varying in the range n=1.3 to n=3.0 (rats) and n=2.0 to n=5.5 (humans). The **top** plots show all the error curves, while the **bottom** plots show magnification of short segment sizes’ errors. One can notice that the fuzzy exponent that gives the best accuracy (lower error) varies according to the segment size, although the optimal exponents are different from those obtained with the original fuzzy function. The lowest error for all segment sizes is illustrated in the plots at the magnification plots’ right (MFE minimum). For reference, MMSE is also illustrated in the plots (gray lines).

**Figure 8 entropy-23-01620-f008:**
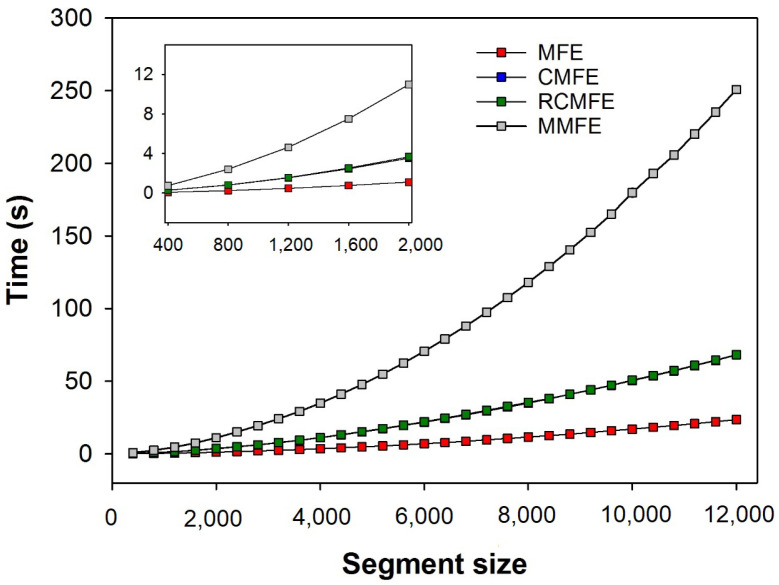
Computation time of the fuzzy entropy-based variants of MSE, i.e., MFE, CMFE, RCMFE, and MMFE. The plot shows the average time consumed (in seconds) to calculate the variants of MSE for 3 HRV series of humans up to segments of 12,000 points. Of note, CMFE and RCMFE take virtually the same time to be computed, and therefore, these curves are superimposed.

## Data Availability

The rat ECG dataset can be requested from the authors. The human ECG dataset is available at https://physionet.org/content/nsr2db/1.0.0/ (accessed on 26 November 2021).

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
