# Peer review of "Multiscale Entropy Analysis of Short Signals: The Robustness of Fuzzy Entropy-Based Variants Compared to Full-Length Long Signals"

_entropy, 2021, doi:10.3390/e23121620_

Round 1
Reviewer 1 Report
Introduction needs to be improved, some more information and details about the aim and objective of the proposed work, along with a presentation of the state of the art regarding multiscale entropy and its variants and their applications should be provided.
- The list of references is quite short, a deeper study on the state of the art, problematic should be provided.
- Eq. 21 should be corrected.
- Line 198 --> In figure 3 there is only one dashed line, hence, singular must be used. Same in figure 2 and figure 3
- Regarding figures, colors are too similar between them, particularly inf figure 3. The use of markers or different line types are suggeted.
- Figure headings should describe the figure not provide results exposition regarding the work. The text regarding the results explanations for each figure, should appear in the results section not here.
- Line 220 --> reference is made to Fig. 6, should this reference point to fig.5 instead?
- lines 233-238, this paragraph describes figure 8, in the last sentence a second reference to fig.8 is given. Not necessary.
- Figure 5 --> CMSE is hidden below RCMSE same to MFE and CMFE with respect to RCMFE and MMFE. One could think this results are not plotted, the use of different markers or line types are highly recommended.
- the author contibutions section is not described
- reference 13 and 14 is the same
Author Response
Introduction needs to be improved, some more information and details about the aim and objective of the proposed work, along with a presentation of the state of the art regarding multiscale entropy and its variants and their applications should be provided.
The authors thank the reviewer for the suggestions. We added sentences detailing and highlighting the objectives.
1. The list of references is quite short, a deeper study on the state of the art, problematic should be provided.
Answer:
We added five recent references reinforcing our arguments and following this suggestion from the reviewer, e.g.:
- Wang, F.; Lu, B.; Kang, X.; Fu, R. Research on driving fatigue alleviation using interesting auditory stimulation based on VMD-MMSE. Entropy 2021, 23, 1209.
- Zheng, J.; Pan, H.; Tong, J.; Liu, Q. Generalized refined composite multiscale fuzzy entropy and multi-cluster feature selection based intelligent fault diagnosis of rolling bearing. ISA transactions 2021.
- Gon¸calves, H.; Henriques-Coelho, T.; Rocha, A.P.; Louren¸co, A.P.; Leite- Moreira, A.; Bernardes, J. Comparison of different methods of heart rate entropy analysis during acute anoxia superimposed on a chronic rat model of pulmonary hypertension. Medical Engineering & Physics 2013, 35, 559–568. doi:10.1016/j.medengphy.2012.06.020.
2. Eq. 21 should be corrected.
Answer:
Equation 21 was corrected in the manuscript accordingly.
3. Line 198 - In figure 3 there is only one dashed line, hence, singular must be used. Same in figure 2 and figure 3.
Answer:
The captions in figures 2 and 3 are corrected.
4. Regarding figures, colors are too similar between them, particularly inf figure
3. The use of markers or different line types are suggested.
Answer:
The authors intended to distinguish between different exponents and indicate the exponents’ numerical tendency through the color gradient. We believe it is important to indicate the numerical trend existing in the exponents of the different MSE patterns.
5. Figure headings should describe the figure not provide results exposition regarding the work. The text regarding the results explanations for each figure, should appear in the results section not here.
Answer:
The authors intend to facilitate readings by briefly describing the results concerning the figure. We believe that summarizing the result on captions can help attract readers’ attention since redundancy is minimized.
6. Line 220 - reference is made to Fig. 6, should this reference point to fig.5 instead?
Answer:
Yes, we mistook the figure number at this point, and the reference is now corrected to figure 5. Thank you for the warning.
7. lines 233-238, this paragraph describes figure 8, in the last sentence a second reference to fig.8 is given. Not necessary.
Answer:
We removed the second reference, the whole parenthesis actually.
8. Figure 5 - CMSE is hidden below RCMSE same to MFE and CMFE with respect to RCMFE and MMFE. One could think this results are not plotted, the use of different markers or line types are highly recommended.
Answer:
The curves have numerical similarities indeed and then are overlapped. We added a sentence on the caption to explain and highlight this fact.
9. the author contibutions section is not described
Answer:
The author would like to thank the reviewer for noticing this issue. We fixed it by including a detailed contributions section.
10. reference 13 and 14 is the same.
Answer:
The author would like to thank the reviewer for noticing this issue again. We fixed it by deleting the duplicated reference.

Reviewer 2 Report
The authors present a study about the use of multiscale entropy analysis with fuzzy entropy-based variants, considering different fuzzy-based approaches. To that purpose, they compared the performance of some methods using a dataset composed of heart rate sequences from humans and rats. The performance was measured as the difference between the fuzzy-based and non-fuzzy (MSE) approaches. I have the following comments to the authors:
1. I am not sure about the appropriateness of the terms “errors” or “accuracy”. If it is the authors’ opinion that MSE is less correct than the fuzzy-based approaches, therefore MSE is not the “gold standard”.
2. Please check the indices in equations (7) and (8).
3. Please provide the sampling frequency of the ECG signals used in the present study.
4. The complexity measured throughout the whole time series may vary depending on the subject/animal and experimental conditions. Therefore, I am not convinced whether the average entropy value of the segments may surely be a kind of estimation of the full series entropy. In order to address this issue, I would recommend the authors to present graphically some of the entropy values throughout all segments for some individuals/animals.
5. Please provide some descriptive measures of the MSE for both the human and animal datasets.
6 . Following my previous comment, depending on the entropy values, mean squared errors of 0.25 (for rats) and 0.12 (for humans) correspond to a difference between MSE and some fuzzy functions of around 0.5 and 0.45, which may be considered quite high. Can the authors elaborate on this?
7. How can the authors explain the huge difference in the mean squared error between the smaller and largest segment sizes? The mean squared error is normalized by the number of points, and so the term 1/N (where N is the number of segments) should not be responsible for such effect, since a greater value of N should also correspond to a greater numerator (number of error sums).
8. What do represent the other curves in the plots of Figure 1-4 with the same behavior as the dashed line, corresponding to the non-fuzzy approach?
9. In addition, can the authors elaborate on the fact that the non-fuzzy and fuzzy approaches present a different behavior as a function of the segment size (fuzzy curves increase as the segment size increases, whereas non-fuzzy decreases)?
10. According with equation (25), the parameter “n” is a function of the segment size. If the number of points of the signals from humans and animals were the same, then why different ranges of values were used for “n” (0.8-1.5 for rats and 0.85-0.92 for humans)?
11. The authors observed that “best exponents increase with the segment size for rats, while it decreases for human HRV series”. However, the range of values for rats is between 0.8 and 1.5, whereas for humans is below 1. With the used colors, I could not distinguish between the curves corresponding to 0.8 and 0.9 for rats in Figures 1-4. This should be carefully addressed (and if the conclusion persists, how can the authors explain it, in addition to “a consequence of the different species”), as well as after addressing also my previous comment concerning the difference in the two ranges of values for “n”.
12. The MFE curve is not visible in Figure 5.
13. The authors studied the application of MSE to an average of around 387 short segments (considering the smaller segment of 400 points), which is not the same as applying MSE to a single segment. Although the present study might still be of interest, it should be clarified throughout the whole manuscript (including title) that the study is focused on short segments of long signals, rather than on short signals. With respect to this, please refer to Medical Engineering & Physics 35 (2013) 559– 568, where MSE was computed up to scale 6, since the analysed segments had 480 points and the minimum number of points required for SampEn computation is 75.
14. The “Data Availability Statement” and other fields at the end of the manuscript are incomplete.
15. References 13 and 14 are the same.
Author Response
The authors present a study about the use of multiscale entropy analysis with fuzzy entropy-based variants, considering different fuzzy-based approaches. To that purpose, they compared the performance of some methods using a dataset composed of heart rate sequences from humans and rats. The performance was measured as the difference between the fuzzy-based and non-fuzzy (MSE) approaches. I have the following comments to the authors:
The authors thank the reviewer for the time spent and for the relevant com- ments.
1. I am not sure about the appropriateness of the terms “errors” or “accuracy”. If it is the authors’ opinion that MSE is less correct than the fuzzy-based ap- proaches, therefore MSE is not the “gold standard”.
Answer:
The study approached the errors and accuracy, considering that reference is the full-length series analyzed by sample entropy in multiple scales (MSE). The authors believe that sample entropy is the measure conceptually closest to the original concept of the estimated KS entropy for deterministic systems or very long series. Therefore, we respectfully disagree with the referee at this point.
2. Please check the indices in equations (7) and (8).
Answer:
We have checked the equation indices, and they are correct and written in agreement with previous publications. We thank anyway for the opportunity to double-check it.
3. Please provide the sampling frequency of the ECG signals used in the present study.
Answer:
The authors modified the dataset subsection to provide the sampling rate information. According to the Physionet MIT-BIH Normal Sinus Rhythm database description, the signals are acquired at a sampling frequency of 360 Hz for each ECG lead, and the animals’ raw data are sampled at 2 kHz.
4. The complexity measured throughout the whole time series may vary de- pending on the subject/animal and experimental conditions. Therefore, I am not convinced whether the average entropy value of the segments may surely be a kind of estimation of the full series entropy. In order to address this issue, I would recommend the authors to present graphically some of the entropy values throughout all segments for some individuals/animals.
Answer:
The complexity measure stationarity is an important issue, and the review is correct that the complexity may vary with the subject and experimental conditions. On the other hand, a graph representing the stationary or non-stationary behavior of the sample entropy would have little meaning in this context, not to mention that it would have to be assessed in the twenty time scales. The calculation of MSE using the whole series is also limited if one considers that the complexity changes at different segments. We believe the average procedure over all segments is the best approximation for the whole-series MSE. There- fore, the graphic representation of entropy over segments would be too dense and perhaps polluted, so we decided not to show these graphs.
5. Please provide some descriptive measures of the MSE for both the human and animal datasets.
Answer:
The total descriptive measures of the MSE for the used dataset are presented in the previous paper from the authors and cited in the manuscript (DOI: 10.1063/5.0010330), so we decided to avoid duplication.
6. Following my previous comment, depending on the entropy values, mean squared errors of 0.25 (for rats) and 0.12 (for humans) correspond to a difference between MSE and some fuzzy functions of around 0.5 and 0.45, which may be considered quite high. Can the authors elaborate on this?
Answer:
The previous study by the authors cited in the manuscript (DOI: 10.1063/5.0010330) shows an exponential relation between the optimum fuzzy parameter and the series length. The MSE is high for shorter series. It adjusts the fuzzy tolerance function to the length; therefore, one can drastically reduce the MSE by using
the optimum fuzzy parameter.
7. How can the authors explain the huge difference in the mean squared error between the smaller and largest segment sizes? The mean squared error is normalized by the number of points, and so the term 1/N (where N is the number of segments) should not be responsible for such effect, since a greater value of N should also correspond to a greater numerator (number of error sums).
Answer:
The mean square errors were calculated concerning each scale factor, comparing scales from 1 to 20 of the original MSE with the 20 scales of each segment’s respective nonfuzzy and fuzzy methods, so the number of scales (N) is always
20. Therefore, we understand that the error discussion should focus on the fuzzy exponent’s dependence on the length of the series.
8. What do represent the other curves in the plots of Figure 1-4 with the same behavior as the dashed line, corresponding to the non-fuzzy approach? Answer:
The other curves in Figure 1-4 have the same behavior as the dashed line, corresponding to the non-diffuse approach. In the respective lower frames on the right, we have a zoom of the upper frame. In the lower frames on the left, we compare the nonfuzzy method with a better exponent for the caption’s fuzzy method.
9. In addition, can the authors elaborate on the fact that the non-fuzzy and fuzzy approaches present a different behavior as a function of the segment size (fuzzy curves increase as the segment size increases, whereas non-fuzzy decreases)?
Answer:
The mean square error varies as a function of the segment size for fuzzy-based approaches depending on the fuzzy exponent adopted. The minimum errors curve for fuzzy-based methods (bottom right plots of the figures) is estimated by selecting the optimal fuzzy exponent for each segment size. On the other hand, the errors for nonfuzzy-based methods always decrease with the segment size. The interesting point here is the curves with optimal fuzzy exponents as a function of the segment size. Those curves are crescent for rats but decreasing for humans, and such findings still need studies for better understanding.
10. According with equation (25), the parameter “n” is a function of the seg- ment size. If the number of points of the signals from humans and animals were the same, then why different ranges of values were used for “n” (0.8-1.5 for rats and 0.85-0.92 for humans)?
Answer:
We apologize for the confusion this equation may have caused. Equation (25) shows the curve fitting for the optimal exponent for MMFE in humans, as previously found in another paper (DOI:10.1063/5.0010330). Notice that it is virtually the same fitting function found here for the optimal exponent of MMFE (Fig. 6). We used this equation to choose the exponents used in the experiment that calculated the computational costs of fuzzy-based methods, and it was per- formed using only series for humans. The fitting functions for rats are shown in Fig. 6. However, as mentioned in the previous answer, it is intriguing that the optimal exponents are so different for humans and rats. Their range and profiles are pretty diverse, possibly pointing to their different intrinsic dynamics.
11. The authors observed that “best exponents increase with the segment size for rats, while it decreases for human HRV series”. However, the range of values for rats is between 0.8 and 1.5, whereas for humans is below 1. With the used colors, I could not distinguish between the curves corresponding to 0.8 and 0.9 for rats in Figures 1-4. This should be carefully addressed (and if the conclusion persists, how can the authors explain it, in addition to “a consequence of the different species”), as well as after addressing also my previous comment concerning the difference in the two ranges of values for “n”.
Answer:
The authors intended to distinguish between different exponents and indicate the exponents’ numerical tendency through the color gradient. We believe it is crucial to show the numerical trend existing in the exponents of the different MSE patterns. Moreover, the different ranges of optimal fuzzy exponents found for humans and rats do not have a clear explanation so far. We believe it may be the consequence of their different dynamics, but it must be addressed in a future dedicated study.
12. The MFE curve is not visible in Figure 5.
Answer:
There are some curves superimposed in the figure including MFE, the figure caption was modified to clarify and explain this fact:
Figure 5 compares errors from all the multiscale variants studied, i.e., CMSE, RCMSE, MMSE, CMFE, RCMFE, MMFE, and MFE. The minimum error is shown for fuzzy entropy-based methods, calculated with each segment size’s optimal fuzzy exponent. The figure shows that all variants based on diffuse entropy have fewer errors than any variant based on sample entropy. For segments sized up to 13000 points, the MFE, CMFE, and RCMFE curves are superimposed because these methods have similar results.
13. The authors studied the application of MSE to an average of around 387 short segments (considering the smaller segment of 400 points), which is not the same as applying MSE to a single segment. Although the present study might still be of interest, it should be clarified throughout the whole manuscript (including title) that the study is focused on short segments of long signals, rather than on short signals. With respect to this, please refer to Medical Engineering & Physics 35 (2013) 559– 568, where MSE was computed up to scale 6, since the analysed segments had 480 points and the minimum number of points required for SampEn computation is 75.
Answer:
The authors thank the referee for raising this discussion. The statistical characteristics of the time series can change if the state of the physiological system is not stationary, as the referee rightly pointed out. The stationarity of the signal is a relevant and complex issue and has to be discussed at some point in the context of the MFE measurement. However, there is no simple way to approach this point without going through issues such as changes in the physi- ological state of the system, as done in the study cited by the referee. We have added a couple of discussion sentences on this point to the manuscript citing the pointed research. Still, we respectfully believe that further discussion is beyond the scope of this study.
14. The “Data Availability Statement” and other fields at the end of the manuscript are incomplete.
Answer:
The authors filled and completed these fields. We thank the referee for this warning.
15. References 13 and 14 are the same.
Answer:
We corrected the duplicated reference. Once again, we thank the referee for noticing.

Reviewer 3 Report
The authors showed that fuzzy-based adaptations of three variants of multiscale entropy calculations for short time series are more accurate than algorithms based on sample entropy. Moreover, accuracy depends on the correct choice of the fuzzy exponent.
The topic of the study is of scientific importance, and the conclusions of the study will have impact on the field of calculation of modified multiscale entropy for short time signals. The manuscript is clear and well written, and the structure is appropriate.
Author Response
The authors showed that fuzzy-based adaptations of three variants of multiscale entropy calculations for short time series are more accurate than algorithms based on sample entropy. Moreover, accuracy depends on the correct choice of the fuzzy exponent.
The topic of the study is of scientific importance, and the conclusions of the study will have impact on the field of calculation of modified multiscale entropy for short time signals. The manuscript is clear and well written, and the structure is appropriate.
We want to thank the referee for the time spent and for the comment.

Round 2
Reviewer 2 Report
The authors addressed the minor issues mentioned in my previous report, but the most important ones were not, in my opinion, adequately addressed, as I detail in the following:
1. With respect to my previous comment #1, as far as I understood the authors’ answer, I respect their point of view, but I would suggest at least two things: 1) clearly state the objective of the work in the abstract and in the introduction; and 2) addressing in the Discussion that a possible limitation of the employed approach was the fact of assuming the same entropy computed for the whole series within shorter segments of it.
2. Regarding my previous comment #4, I was not suggesting to present a full report, but rather some of the entropy values throughout all segments for some individuals/animals, which can be included as an Appendix. I believe this would be of interest for the readers and enrich the manuscript, supporting the adopted approach of averaging.
3. I would suggest at least mentioning explicitly in the Results that the descriptive measures of the MSE for both the human and animal datasets can be found in a previous paper from the authors.
4. In my opinion, the authors’ answer to my previous comments #6 and #7 were not sufficiently clear. Additionally, the corresponding discussion should be added to the manuscript.
5. The authors’ answer to my previous comment #10 should be accomplished by corresponding changes in the manuscript to clarify this issue.
6. With respect to my previous comments #8 and #11, I am not still convinced that the adopted colors/symbols provide a sufficient understanding of the presented results.
7 .I could not identify any change in the caption of Figure 5, as the authors stated in their answer to my previous comment #12.
8. Regarding my previous comment #13, I am still of the opinion that the manuscript in the present form might induce the readers in error, since the analysis of a short signal is different from the analysis of an average of hundreds of short segments from a long signal.
9. There are still sections at the end of the manuscript, namely “Institutional review board statement”, “Informed consent statement” and “Data availability statement”, which are still incomplete.
Author Response
Please see the attached response letter.

Round 3
Reviewer 2 Report
The authors satisfactorily addressed most of my previous comments, but there are still in my opinion some pending issues:
- With respect to the authors’ answer to my previous comment #1, the title should be changed accordingly.
- A reference to the introduced Appendix A should be added to the manuscript, as well as mentioning whether despite a single case is presented, it can be considered a representative example of the whole dataset.
- Regarding the authors’ answer to my previous comment #4, there are other figures than figure 5 where MSE is in fact around 0.25. Can the authors elaborate on this? With respect to the second part of their answer, I would suggest adding some comment to the paper.
Author Response
1. With respect to the authors’ answer to my previous comment #1, the title should be changed accordingly.
Answer:
We changed the title to Multiscale Entropy Analysis of Short Signals: the Robustness of Fuzzy Entropy-Based Variants compared to full-length long signals.
2. A reference to the introduced Appendix A should be added to the manuscript, as well as mentioning whether despite a single case is presented, it can be considered a representative example of the whole dataset.
Answer:
We thank and agreed with the referee, adding one sentence to the discussion and the appendix. The sentence in the Discussion section (second paragraph) is “We introduce an illustrative analysis of two time-series data, i.e., one human and one rat, in the appendix to this paper presented in Fig. A1”. Moreover, the sentence included in the Appendix section is “The chosen subject and animal are typical and representative of the whole datasets for both human and rats respectively..”
3. Regarding the authors’ answer to my previous comment #4, there are other figures than figure 5 where MSE is in fact around 0.25. Can the authors elaborate on this? With respect to the second part of their answer, I would suggest adding some comment to the paper.
Answer:
There is a Mean Squared Error (MSE) around 0.25 for inadequate Fuzzy exponent for that length indeed. That’s why we propose the Fuzzy exponent given in Figure 6. If one uses the indicated Fuzzy exponent, the MSE (errors) is around 0.001 for rats and humans, respectively.
We added the following discussion on the text (Experiments): The mean squared error is obtained by calculating entropy for time scales from 1 to 20 for each segment of 400, 800,..., 15600 points. If the number of windows is greater than 1, the arithmetic average of each scaling factor is made. For example: for the 400-point segment, we have 385 windows. For each window, we calculated entropy on the 20 time scales. Then, the arithmetic mean of the 385 windows is taken for each scaling factor, so the entropy for scale 1 and the 400-point segment is the entropy average for scale 1 of all 385 windows. The entropy for scale 2 and the segment of 400 points is the average entropy on scale 2 of all 385 windows, so up to scale 20. As a result, we have an average entropy for scales 1, 2, …, 20. We then average the mean squared error of these entropy values with the calculated MSE entropy on a scale of 1 to 20.
